# ReSafe: Enhancing Safety of Text-to-Image Diffusion via Post-Hoc Image Back Translation

## Abstract

Ensuring safe images in Text-to-Image (T2I) diffusion models has emerged as an active area of research. However, existing T2I safe image generation methods may fail to fully erase learned knowledge and remain vulnerable to circumvention like adversarial prompts or concept arithmetic. Given that safe image generation methods can be bypassed, we introduce a post-hoc approach designed to uphold safety even in the presence of such circumvention. We present **ReSafe**, the first Image-to-Image (I2I) translation framework designed to regenerate safe images from unsafe inputs by removing only harmful features while preserving safe visual information. ReSafe extracts safe multimodal (i.e., vision and language) features by selectively removing unsafe concepts from the input representations. It then optimizes a discrete safe prompt to align with the interpolated multimodal safe features and generates new safe images from this prompt, effectively eliminating unsafe content while preserving semantic and visual information. Since ReSafe is a post-hoc approach, it can be applied to a variety of existing safe image generation methods to enhance their performance. ReSafe reduces attack success rates by 3–4× compared to T2I methods and by 3–7× compared to I2I baselines across five adversarial prompt benchmarks.

*Warning: This paper includes examples of harmful language and images that may be sensitive or uncomfortable. Reader discretion is advised.*

## 1 Introduction

Recent advances in Text-to-Image (T2I) generation models (Rombach et al., 2022; Saharia et al., 2022; Podell et al., 2024; Esser et al., 2024; Labs, 2024), particularly diffusion models (Sohl-Dickstein et al., 2015; Song & Ermon, 2019; Ho et al., 2020), have significantly improved the fidelity and diversity of synthesized images. Ensuring the safety of generated content has accordingly become a critical research direction in this field. Many researchers have focused on controlling the generation process based on input prompt, aiming to guide the models toward safe outputs through techniques such as concept erasing (Gandikota et al., 2023a; Kumari et al., 2023; Lu et al., 2024; Huang et al., 2023; Fan et al., 2023; Lee et al., 2025; Li et al., 2025), weight modification (Gandikota et al., 2023b; Gong et al., 2024), or training-free methods (Yoon et al., 2025; Schramowski et al., 2022a). Such methods are typically designed to eliminate unsafe content, including depictions of nudity, violence, and even certain artistic styles.

Existing safe image generation methods have primarily focused on T2I models that generate safe images given a textual prompt. These approaches typically operate at the text encoder level (Radford et al., 2021), either by fine-tuning the model to produce alternative safe images (Kumari et al., 2023; Gandikota et al., 2023a) for specific unsafe prompts or by redirecting them through safer alternative prompts (Yoon et al., 2025), as illustrated in Fig. 1 (a). Recent studies indicate that unlearned models tend to conceal rather than forget knowledge (Sharma et al., 2024), and remain vulnerable to adversarial prompts (Tsai et al., 2024; Zhang et al., 2024b) and concept arithmetic attacks (Petsiuk & Saenko, 2024), which can easily bypass the defense and lead to the generation of unsafe images (see Fig. 1 (b)). Therefore, given that unlearning models may still retain target unlearning concepts or remain vulnerable to attacks, it is important to consider additional post-hoc safety approaches to ensure safe image generation. This motivates our research question: **How can we regenerate a safe image from an unsafe input image by selectively removing the unsafe attributes or concepts, while preserving the remaining semantic and visual information?**

Inspired by Image-to-Image (I2I) translation approaches (Liu et al., 2017; Brooks et al., 2023; Chen et al., 2025; Zhang et al., 2025), we aim to selectively remove only the unsafe attributes or concepts from an unsafe image and transform it into a safe counterpart. Although recent I2I models such as InstructPix2Pix (Brooks et al., 2023) and Instruct-CLIP (Chen et al., 2025) show strong performance on general edits, they cannot be naively applied to address unlearning instructions, such as transforming an unsafe image into a safe one. This is because conventional I2I focuses on concrete, object-centric, or explicit visual modifications (e.g., adding/removing objects) rather than abstract or semantic-level transformations like safety adjustments, largely due to the lack of annotated image pairs mapping unsafe inputs to their safe counterparts. Thus, we propose a novel I2I based approach that enables safe image regeneration from the unsafe inputs, specifically designed to support unlearning objectives even in the absence of unsafe–safe paired datasets.

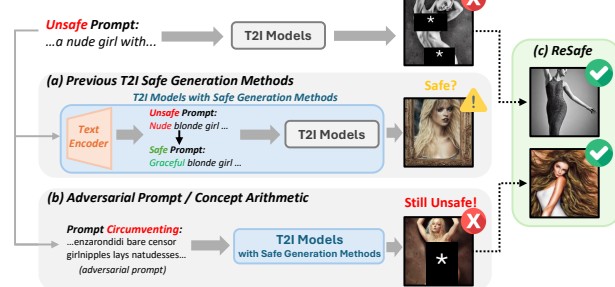

Figure 1: We present ReSafe, a novel I2I framework that removes inappropriate features from an image to a safe counterpart. (a) Previous T2I safe image generation methods remain vulnerable to (b) Adversarial prompts and concept arithmetic attacks. In contrast, (c) ReSafe performs post-hoc translation of unsafe images into safe ones while maintaining the original semantic information. We use ■ for publication purposes.

To regenerate a safe image from an unsafe input, one straightforward approach is simply prompting the Vision-Language Model (VLM) to rewrite an unsafe image caption to be safe, and then using this revised caption as input to Stable Diffusion to synthesize the safe image. However, prior studies on prompt optimization (Wen et al., 2023; Mahajan et al., 2024; Kim et al.) have demonstrated that captions produced directly by VLMs often fail to capture the fine-grained semantic features of target images. Thus, using optimized prompts is generally more effective for synthesizing images that faithfully reflect the desired attributes (see Appendix B.1 for further details and experimental results). Since our I2I framework aims to extract and preserve safe features from the input image, it is natural to adopt prompt optimization rather than relying solely on a simple caption.

We introduce **ReSafe**, the first and novel I2I translation framework for safety, which regenerates a safe image from an unsafe input by removing only the unsafe features while preserving other safe information obtained through prompt optimization. To effectively enable unsafe-to-safe transformation in I2I tasks, we first extract a safe image feature from the unsafe input. However, relying solely on the image feature with prompt optimization (Wen et al., 2023) can result in information loss during regeneration (see Fig. 6 (a)). To address this issue, we apply interpolated prompt optimization during the image-to-text process, leveraging multimodal (i.e., visual and textual) features to minimize information loss between the input and the generated output. Since our final goal is to transform unsafe images into safe ones, we extract both safe visual and safe textual features from the unsafe input through the safe features extraction process. By interpolating the extracted safe multimodal features and applying prompt optimization, we obtain a discrete prompt that effectively encodes the safe semantics of the original unsafe image. This prompt can then be used with various diffusion models (e.g., SDXL (Podell et al., 2024), SDv3 (Esser et al., 2024) and Flux (Labs, 2024)) to regenerate safe images in which the unsafe features present in the input image are removed.

ReSafe is the first safety-aware I2I translation framework and can be integrated with existing T2I-based safety methods. Unlike the default safety filters in Stable Diffusion (SD) (Rombach et al., 2022), which often return black images in response to an unsafe prompt, ReSafe retains the utility of safe content present in the input. Compared to recent I2I translation models, ReSafe reduces the generation of unsafe content by more than 3×, while successfully regenerating safe images. Furthermore, ReSafe demonstrates significantly improved performance across five adversarial prompt benchmarks (Tsai et al., 2024; Yang et al., 2024; Chin et al., 2024; Zhang et al., 2024b; Schramowski et al., 2022a) when combined with concept erasure, weight modification, and training-free methods. Our comprehensive experiments, including quantitative and qualitative results, ablation studies, and a thorough analysis of the ReSafe framework, highlight the effectiveness of our method as a safety mechanism and position it as a foundation for a new paradigm in safety-aware I2I translation.

Our key contributions can be summarized as follows:

- We introduce ReSafe, the first safety-aware I2I translation method that regenerates safe images from unsafe inputs by removing only the unsafe features while preserving other safe semantic and visual information.

- For effective unsafe-to-safe image translation, we extract both visual and textual safe features from the unsafe input using our safe features extraction. ReSafe then interpolates these features for discrete prompt optimization that guides the generation of images containing only the safe aspects of the input image.

- As a post-hoc approach, ReSafe is compatible with existing unlearning methods and can be integrated alongside them. ReSafe achieves state-of-the-art performance on five adversarial benchmarks, both when applied to existing I2I models and when combined with T2I-based safety methods, providing strong evidence that our framework introduces a new paradigm for safe I2I translation.

## 2 PRELIMINARY

We begin with preliminaries by introducing the two key components necessary for understanding ReSafe: (1) Prompt optimization, and (2) Image Back Translation, which is inspired by back translation in natural language processing (NLP).

**Prompt optimization.** Prompt optimization (Pryzant et al., 2023; Wen et al., 2023; Kim et al.) learns a sequence of discrete tokens (hard prompt) whose text features closely align with the visual features of a given image. Let $\mathbf{v} \in \mathbb{R}^d$ denote the CLIP (Radford et al., 2021) visual embedding of an input image, and let $\mathcal{V} = \{t_1, \ldots, t_N\} \in \mathbb{R}^d$ be the set of pre-trained token embeddings in the text vocabulary, where $\mathbb{R}^d$ is the $d$-dimensional embedding space. PEZ optimizes a continuous prompt embedding $\mathbf{z} \in \mathbb{R}^{k \times d}$, where k is the number of tokens, iteratively. During training, each row $\mathbf{z}_i$ is projected onto the closest token in $\mathcal{V}$ via:

$$\hat{t}_i = \arg\min_{t \in \mathcal{V}} ||z_i - t||_2 \tag{1}$$

The sequence $\hat{\mathbf{T}} = [\hat{t}_1, ..., \hat{t}_k]$ is then passed through the CLIP text encoder to obtain the prompt feature $f_T \in \mathbb{R}^d$. The objective is to maximize the cosine similarity between $f_T$ and the visual embedding $\mathbf{v}$:

$$L_{cos} = 1 - \frac{f_T^\top \mathbf{v}}{||f_T|| \cdot ||\mathbf{v}||} \tag{2}$$

By minimizing $L_{cos}$, the prompt embedding is gradually aligned with the semantics of the input image. After training, the resulting discrete tokens $\hat{\mathbf{T}}$ can be used not only to regenerate images that are semantically similar to the original input, but also in various applications such as style transfer, compositional concept synthesis, and prompt distillation.

**Back Translation and Image Back Translation.** Back Translation (Sennrich et al., 2016; Corbeil & Ghadivel, 2020) is a data augmentation technique in conventional NLP. Given a source language text $T_{source}$, the method translates it into a target language to obtain $T_{target}$, and subsequently back-translates it into the source language to produce $T'_{source}$. This process yields paraphrased text that preserves the original semantics while introducing lexical or syntactic variation in surface form.

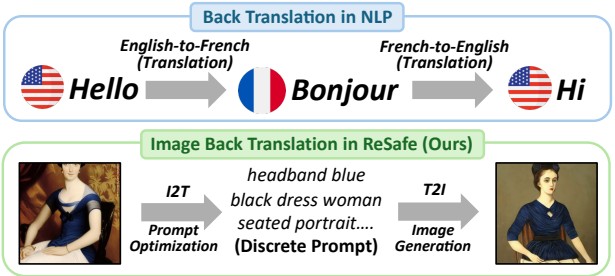

Figure 2: Image Back Translation in ReSafe inspired by back translation in NLP

For example, as shown in Fig. 2, an English input such as "Hello" ($T_{source}$) can be translated to the French word "Bonjour" ($T_{target}$) using an English-to-French translator. When "Bonjour" is then translated back into English, it may produce a semantically equivalent yet syntactically distinct phrase like "Hi" ($T'_{source}$). Such back-translated outputs introduce natural linguistic variations into the dataset, which can be effectively leveraged to enhance model robustness and generalization.

Inspired by this paradigm, we propose Image Back Translation, which performs translation across image and text modalities, specifically in an image to text to image manner. Given an input image $I_{before}$, we first convert it into an optimized textual prompt $T_{optim}$ via prompt optimization. This optimized prompt is then used to generate a new image $I_{after}$ that preserves the essential semantics and visual content of $I_{before}$. In this paper, we reinterpret Image Back Translation not as a data augmentation technique, but as a novel I2I translation framework for converting unsafe images into safe ones. For more discussion about Image Back Translation, please refer to the Appendix C.

## 3 ReSafe

ReSafe is to transform an unsafe image—whether originally generated or externally provided—into a safe version by selectively removing harmful concepts or attributes, while preserving the original visual semantics. ReSafe consist of the three steps: (1) safe features extraction from unsafe images, introduced in Sec.3.1; (2) interpolating multimodal safe features for safety-semantic feature, described in Sec.3.2; and (3) regeneration of safe images using the optimized prompt, completing the ReSafe pipeline in Sec. 3.3. The ReSafe framework is illustrated in Fig. 3

### 3.1 Safe Features Extraction from an Unsafe Image

Since our goal is to generate a safe image from an unsafe input, we first need to extract the safe features from the unsafe image. To better preserve the safe information contained in the input image, we extract safe features from both visual and textual modalities. In this section, we describe how to extract both safe text features and safe image features from an unsafe input image.

**Safe text feature extraction.** To obtain the safe text feature $f_{txt}^{safe}$ from the input (unsafe) image, we utilize Vision Language Model (VLM) (Wang et al., 2024a) to generate safe and unsafe text captions. As illustrated in Fig. 3 (b), the original unsafe image is passed through the VLM to generate a caption that semantically describes its visual content, which we refer to as the unsafe caption $f_{txt}^{unsafe}$. In here, we use the prompt "Describe this image in a caption." to query the VLM. Next, we identify specific unsafe components within the caption, such as "nude" as illustrated in Fig. 3. These components are extracted as a structured list based on predefined semantic categories (nudity, violence, artistic styles). Using this list, we then prompt the VLM to regenerate a caption that excludes the identified unsafe elements while preserving the remaining safe content. The resulting safe caption is used to extract the corresponding safe text feature $f_{txt}^{safe}$ that captures the safe semantics of the input image without containing harmful or unsafe features. Additional details on the caption extraction process, including the prompts used with the VLM, are provided in the Appendix.

**Safe image feature extraction.** Since the input image inherently contains unsafe visual content, naively encoding it with an image encoder may produce an image feature $f_{img}^{unsafe}$ that also contains undesirable or harmful semantics. To derive safe image features $f_{img}^{safe}$ from $f_{img}^{unsafe}$, we leverage the fact that our CLIP encoders operate in a joint image-text embedding space, along with the previously extracted text-based safe ($f_{txt}^{safe}$) and unsafe ($f_{txt}^{unsafe}$) features. Given unsafe text feature $f_{txt}^{unsafe}$ and safe text feature $f_{txt}^{safe}$, we can define the semantic correction direction as:

$$\Delta_{safe} = f_{txt}^{safe} - f_{txt}^{unsafe} \tag{3}$$

We then project the unsafe image feature $f_{txt}^{unsafe}$ onto this direction to obtain its unsafe component $\mathbf{c}_{unsafe}$ via orthogonal projection:

$$\mathbf{c}_{unsafe} = \frac{\left\langle f_{img}^{unsafe}, \Delta_{safe} \right\rangle}{\|\Delta_{safe}\|^2} \Delta_{safe} . \tag{4}$$

With unsafe component $\mathbf{c}_{unsafe}$ extracted from $f_{txt}^{unsafe}$, the safe image feature $f_{txt}^{safe}$ is then derived from $f_{txt}^{unsafe}$ as follows:

$$f_{img}^{safe} = f_{img}^{unsafe} - \lambda_{unsafe}\mathbf{c}_{unsafe}. \tag{5}$$

where $\lambda_{unsafe}$ denotes the removal scale of unsafe components.

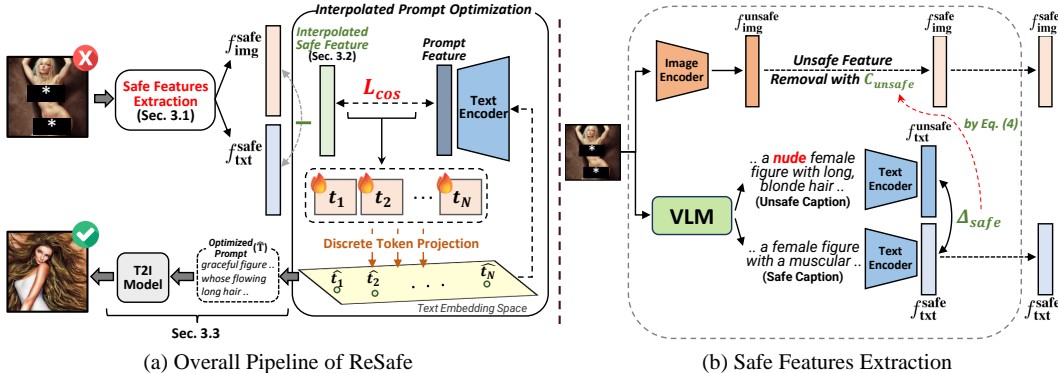

(a) Overall Pipeline of ReSafe

(b) Safe Features Extraction

Figure 3: **(a) The ReSafe framework.** ReSafe extracts and interpolates the multimodal safe features. Then the discrete prompt is optimized with interpolated safe features. And finally, we can regenerate a safe image with the optimized prompt. **(b) Illustration of Safe Features Extraction.** In Safe Feature Extraction, unsafe concept components are removed from both image and text embeddings, and the resulting representations are combined into a unified safe feature space.

This operation subtracts the component of $f_{img}^{unsafe}$ aligned with the unsafe-to-safe direction, effectively pushing the image feature away from unsafe semantics. The resulting features $f_{img}^{safe}$ and $f_{txt}^{safe}$ are used as the ground-truth target during intermediate discrete prompt optimization.

## 3.2 INTERPOLATING MULTIMODAL SAFE FEATURES FOR SAFETY-SEMANTIC FEATURE

One of our key observations is that when Image Back Translation relies solely on image feature based prompt optimization (such as PEZ), the regenerated image often fails to faithfully capture the original content at both perceptual and semantic levels (see Fig. 6 (a)). This is because, although CLIP (Radford et al., 2021) is trained in a joint image–text embedding space, simply maximizing image–text similarity does not guarantee that the optimized prompt fully captures the original image's rich visual details.

To faithfully reconstruct an image $I_{after}$ that is semantically and visually aligned with the original input image $I_{before}$, it is essential to preserve both low-level visual features and high-level semantic content during the back translation process. Relying solely on either image features or text features can lead to the omission of core content or loss of critical information, as discussed in the previous section. To address this issue, we propose an Interpolated Prompt Optimization that leverages both the image features $f_{img}$ of the image and the text feature $f_{txt}$ of its textual description. Based on the fact that CLIP space is a joint embedding space for both images and text, and that the image feature $f_{img}$ is extracted from the CLIP image encoder, we construct an interpolated representation $f_{inter}$ via Spherical Linear Interpolation (Slerp) (Shoemake, 1985), defined as:

$$f_{inter} = \frac{sin((1-\alpha)\theta)}{sin(\theta)} f_{img} + \frac{sin(\alpha\theta)}{sin(\theta)} f_{txt} \qquad (6)$$

where $\theta = \arccos\left(\frac{f_{img} f_{txt}}{||f_{img}||\cdot||f_{txt}||}\right)$ and $\alpha \in [0, 1]$ controls the interpolation ratio.

This interpolated feature $f_{inter}$ combines both visual fidelity and semantic richness, and is used as the ground truth for the prompt optimization.

## 3.3 REGENERATING A SAFE IMAGE FROM OPTIMIZED SAFE PROMPT

Once the safe image feature $f_{img}^{safe}$ and the safe text feature $f_{txt}^{safe}$ have been obtained, we compute an interpolated representation $f_{inter}^{safe}$ using the Slerp formulation described in Eq. 6. This interpolated vector captures a smooth semantic blend of both visual and textual safety-aligned information. We then optimize discrete prompt $\hat{T}$ such that its CLIP text embedding maximizes the cosine similarity with $f_{inter}^{safe}$. As a result, the learned prompt encodes only the safe components of the original unsafe image, having effectively removed any harmful or undesired features during the interpolation and

Table 1: Comparison of ASR between training-free and training-based T2I safe image generation methods. Due to space limitations, we compare ReSafe with representative state-of-the-art baselines from each category. **We provide additional comparisons in the Appendix B.3.**

| Method | No Weights Modification | Training-Free | Ring-A-Bell | | | MMA-Diffusion↓ | P4D↓ | UnLearnDiffAtk↓ | I2P↓ |
|---|---|---|---|---|---|---|---|---|---|
| | | | K77↓ | K38↓ | K16↓ | | | | |
| SDv1.4 (Rombach et al., 2022) | - | - | 95.79 | 97.89 | 98.94 | 95.60 | 92.65 | 68.31 | 18.88 |
| + ReSafe (Ours) | | | **29.47** | **22.10** | **26.32** | **32.70** | **32.35** | **16.19** | **4.40** |
| GLoCE (Lee et al., 2025) | ✗ | ✗ | 4.21 | 3.16 | 1.05 | 1.10 | 6.25 | 5.63 | 3.57 |
| + ReSafe (Ours) | | | **0.00** | **0.02** | **0.00** | **0.20** | **0.00** | **0.00** | **0.00** |
| RECE (Gong et al., 2024) | ✗ | ✓ | 7.37 | 12.63 | 11.58 | 57.90 | 39.71 | 28.17 | 6.4 |
| + ReSafe (Ours) | | | **1.05** | **3.16** | **3.16** | **9.15** | **17.8** | **8.82** | **1.53** |
| SAFREE (Yoon et al., 2025) | ✓ | ✓ | 35.79 | 45.26 | 57.89 | 63.40 | 48.16 | 25.35 | 4.10 |
| + ReSafe (Ours) | | | **16.84** | **14.74** | **10.53** | **8.45** | **19.8** | **18.38** | **0.87** |

Table 2: Comparison of ASR across I2I generation methods.

| Method | Ring-A-Bell | | | MMA-Diffusion↓ | P4D↓ | UnLearn-DiffAtk↓ | I2P↓ |
|---|---|---|---|---|---|---|---|
| | K77↓ | K38↓ | K16↓ | | | | |
| InstructPix2Pix (Brooks et al., 2023) | 91.58 | 91.58 | 97.89 | 94.80 | 89.71 | 60.57 | 15.84 |
| Instruct-CLIP (Chen et al., 2025) | 91.58 | 93.68 | 96.84 | 94.80 | 89.34 | 61.97 | 15.88 |
| ICEdit (Zhang et al., 2025) | 92.63 | 94.74 | 98.95 | 94.70 | 90.07 | 61.27 | 15.48 |
| ReSafe (Ours) | **29.47** | **22.10** | **26.32** | **32.70** | **32.35** | **16.19** | **4.40** |

projection steps. In the final T2I stage, the optimized prompt $\hat{\mathbf{T}}$ is used as input to a standard T2I diffusion model. Notably, our approach does not require any modification or fine-tuning of the diffusion model itself. This means that it avoids the performance degradation often observed in prior unlearning-based methods and remains fully compatible with a wide range of state-of-the-art diffusion models, including SDXL (Podell et al., 2024), SDv3 (Esser et al., 2024), and FLUX (Labs, 2024). In the following section, we present experimental results that demonstrate the effectiveness of our proposed method.

## 4 EXPERIMENTS

### 4.1 BASELINES AND EXPERIMENTAL SETUP

**Baselines.** Although ReSafe is a form of I2I translation, we compare ReSafe not only with the state-of-the-art I2I models (Brooks et al., 2023; Chen et al., 2025; Zhang et al., 2025) but also with existing T2I safe image generation methods, including concept erasing (Kumari et al., 2023; Gandikota et al., 2023a; Lu et al., 2024; Huang et al., 2023; Fan et al., 2023; Lee et al., 2025; Li et al., 2025), weight modification (Gandikota et al., 2023b; Gong et al., 2024), and training-free (Schramowski et al., 2022a; Yoon et al., 2025). Since ReSafe requires an input image, we use unsafe images generated by SDv1.4 when evaluating our model. For both quantitative and qualitative evaluations, we assessed the removal of nudity, violence, and artist-specific concepts, respectively. We employ five adversarial prompt benchmarks to generate unsafe images for red-teaming methods, including Ring-A-Bell (Tsai et al., 2024), MMA-Diffusion (Yang et al., 2024), P4D (Chin et al., 2024), UnLearnDiffAtk (Zhang et al., 2024b), and I2P (Schramowski et al., 2022a).

**Experimental setup.** We set the removal scale $\lambda_{\text{unsafe}}$ to 1, the interpolation ratio $\alpha$ to 0.7, and the number of discrete tokens $N$ to 16. Ablation results are reported in Sec. 4.4, and detailed experimental settings, including learning rates, optimizer, and additional ablation studies, are provided in the Appendix A. All experiments are conducted on a single NVIDIA RTX 4090 GPU.

### 4.2 QUANTITATIVE EVALUATION

For nudity removal, we compute the Attack Success Rate (ASR) using NudeNet (notAI tech, 2019) by counting the proportion of generated images whose predicted nude class probability exceeds 0.45. Tab. 1 presents the ASR results of applying ReSafe to the state-of-the-art T2I safe methods from each category. Note that ReSafe does not require fine-tuning, preserving the original generative capabilities of the diffusion model, the same advantage as in training-free methods. Tab. 2 presents

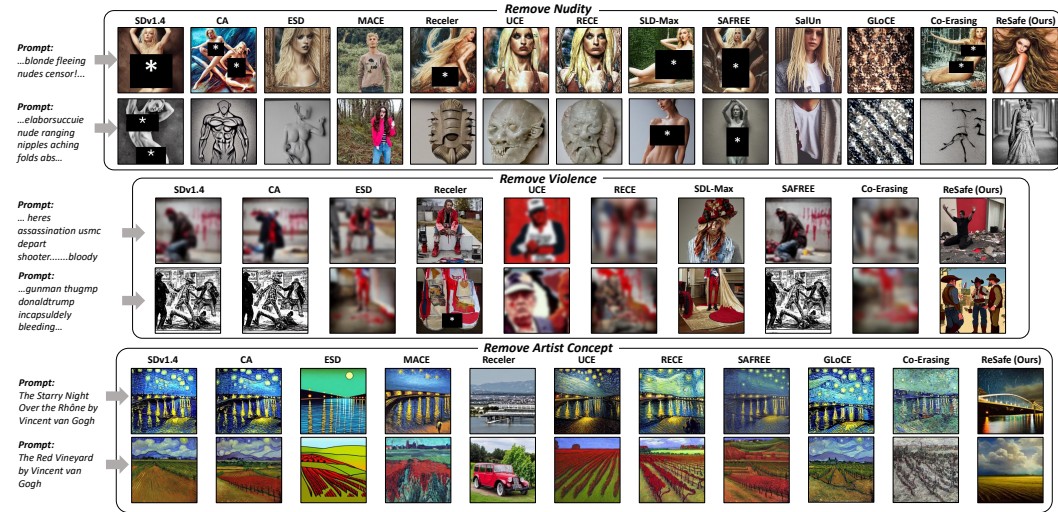

Figure 4: Qualitative comparisons of ReSafe and T2I safe image generation methods. Considering that T2I takes a text prompt as input, resulting in highly varied images, ReSafe demonstrates the ability to generate much safer images compared to existing baselines. More examples are in the Appendix B.2.

Table 3: Comparison across 400 samples of image and text similarity between original images and those generated by Interpolated Prompt Optimization and Prompt Optimization (PEZ).

| Method | Image Similarity↑ (DINOv2) | Image Similarity↑ (CLIP) | LPIPS↓ | Text Similarity↑ (CLIP) | Text Similarity↑ (GPT-Judge) |
|---|---|---|---|---|---|
| Prompt Optimization | 0.5320 | 0.7227 | 0.8033 | 0.6128 | 0.0625 |
| Interpolated Prompt Optimization (Ours) | **0.6003** | **0.7665** | **0.7922** | **0.7400** | **0.9375** |

ASR results from recent I2I models given an instruction prompt such as "Make this image safe."(see Appendix D.1 for details), reflecting the superiority of ReSafe as an I2I approach. Since existing I2I models target object-centric editing rather than safety, their ASR remains extremely high. In contrast, ReSafe substantially reduces ASR, demonstrating that it not only enforces safety much more effectively but also enables generalization to novel erasing concepts compared to previous I2I models. Additional results for nudity and quantitative results for violence and artistic removal are provided in the Appendix B.3.

### 4.3 QUALITATIVE EVALUATION

Fig. 4 illustrates the results of ReSafe in comparison to existing T2I safe image generation methods. Although ReSafe is applied to unsafe images generated by Stable Diffusion v1.4, it achieves performance on par with existing T2I safe image generation methods. Moreover, when combined with more recent safety frameworks (e.g., MACE or SAFREE), the results further improve, as shown quantitatively in Tab. 5. Additional qualitative examples are provided in the Appendix B.2.

Fig. 5 shows the results of applying existing I2I models to unsafe images with an instruction prompt mentioned above. Since such models are trained without unsafe-safe supervision and rely only on general image editing data, they fail to handle more abstract or complex instructions related to safety. As a result, they often produce outputs that are nearly identical to the original unsafe images. In contrast, ReSafe removes the targeted unsafe features (as well as artistic concepts) and regenerates the image while preserving the safe information of the original, even in the absence of an unsafe–safe paired dataset. This supports our view that the proposed Image Back Translation establishes a new paradigm within the advanced I2I translation framework for the safety of image generation.

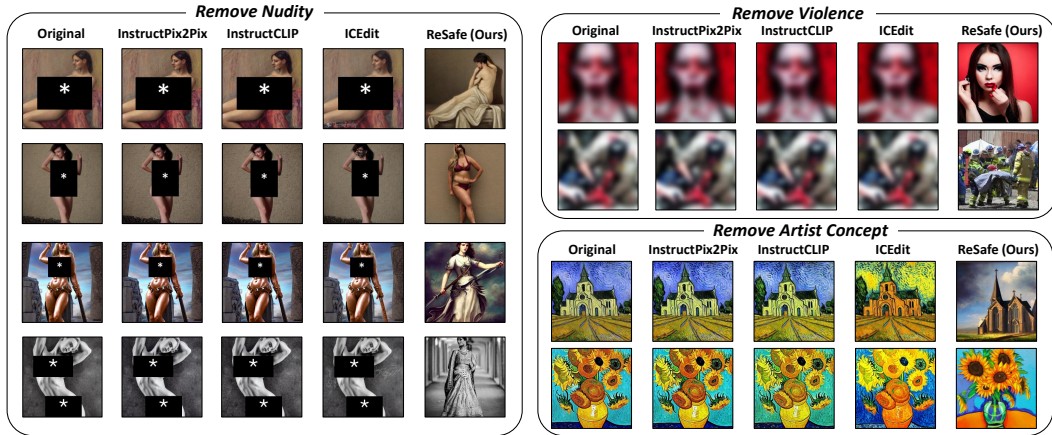

Figure 5: Qualitative comparisons of ReSafe and I2I baselines. ReSafe generates safer images compared to baselines. We use the prompts "Make this image safe." for Nudity and Violence, and "Remove any stylistic influence of {artist} from the image." for Styles.

### 4.4 FURTHER ANALYSIS

**Ablation studies.** Fig. 6 presents an ablation study of our component, Interpolated Prompt Optimization. The image-only variant corresponds to the case where $\alpha = 0$, meaning that only the safe image features are used for prompt optimization, while the text-only variant uses only text features with $\alpha = 1$. In all experiments, we empirically set $\alpha = 0.7$ as the default. Further ablation studies on the choice of $\alpha$, as well as additional experiments on other hyperparameters such as prompt length $N$ and removal scale $\lambda_{\text{unsafe}}$, are provided in the Appendix B.5.

**Does the Image Back Translation process preserve information well?** One may question whether the input and output images in Image Back Translation truly remain the same. However, similar to Back Translation in NLP, where the original and translated texts are not structurally identical but preserve the core meaning, our approach focuses on maintaining the essential semantics of the original image rather than achieving pixel-level identity. As shown in Fig. 6

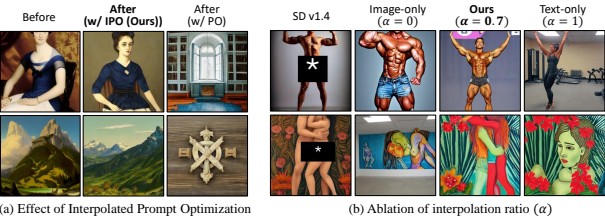

(a) Effect of Interpolated Prompt Optimization     (b) Ablation of interpolation ratio ($\alpha$)

Figure 6: (a) Comparison between Interpolated Prompt Optimization (Ours) and Prompt Optimization. (b) Ablation study on the interpolated ratio ($\alpha$).

(a), the input and output images exhibit strong structural similarity. This observation is further supported by the quantitative metrics reported in Tab. 3. In the table, Image Similarity is measured by the cosine similarity of image embeddings using DINOv2 (Oquab et al., 2023) and CLIP (Radford et al., 2021). Text Similarity is measured by the cosine similarity of text embeddings using CLIP between captions generated by Qwen2-VL (Wang et al., 2024a) for the original and generated images. We also evaluate with GPT-Judge, defined as the percentage of GPT-4o pairwise comparisons in which a method's generated caption is judged more faithful to the original. Notably, Interpolated Prompt Optimization yields higher similarity compared to vanilla prompt optimization, as it leverages both visual and texture features more effectively. These results suggest that applying Image Back Translation with our proposed Interpolated Prompt Optimization leads to output images that better preserve the essential information encoded in the extracted safe features.

**Applications with various diffusion models.** ReSafe transforms the interpolated safe features into an interpretable discrete prompt during the prompt optimization process. This allows the optimized prompt to be used not only with SDv1.4, but also with a wider range of other generative models such as SDXL, SDv3, and Flux. Fig. 7 shows the results of using the same

optimized prompt across various T2I models, demonstrating that our approach can produce safe images of varying styles and improved quality. This suggests that ReSafe is compatible with diverse generative models and can be generalized to broader applications. We provide quantitative evaluation results for various T2I models in the Appendix B.3.

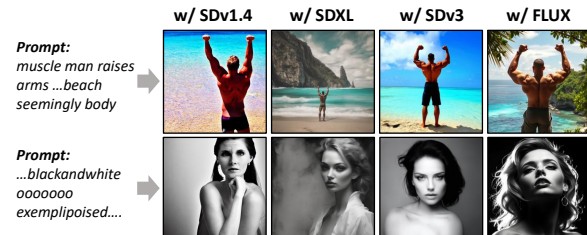

Figure 7: ReSafe with various T2I models.

## 5 RELATED WORK

**Safe image generation.** Ensuring safe image generation has been addressed in previous studies through methods such as concept erasing (Gandikota et al., 2023a; Kumari et al., 2023; Heng & Soh, 2023; Lu et al., 2024; Lyu et al., 2024; Huang et al., 2023; Park et al., 2024; Zhang et al., 2024a; Fan et al., 2023; Bui et al., 2025; Lee et al., 2025; Li et al., 2025), weight modification (Gandikota et al., 2023b; Gong et al., 2024), and training-free approaches (Schramowski et al., 2022a; Yoon et al., 2025). Traditional concept erasing methods involve fine-tuning the diffusion model to suppress undesired concepts, guiding it to produce alternative outputs instead. Nevertheless, this process is computationally expensive and may compromise the model's original generative quality. While weight modification and training-free methods alleviate the limitations of fine-tuning, they are still restricted to T2I models and remain vulnerable to attacks such as adversarial prompts (Tsai et al., 2024; Yang et al., 2024; Zhang et al., 2024b; Chin et al., 2024; Schramowski et al., 2022a) and concept arithmetic (Petsiuk & Saenko, 2024). In contrast, our work presents the first safety-aware I2I framework that regenerates images by preserving the safe features of already generated unsafe images while removing its unsafe features.

**Prompt optimization.** Prompt optimization (Wen et al., 2023; Wang et al., 2024b; Jia et al., 2022) is a widely used technique not only in image generation models (Wang et al., 2024b; Gal et al., 2022) but also in Large Language Models (Zhu et al., 2024) and Vision Language Models (Yao et al., 2023). In image generation, prompt optimization methods can be categorized into soft and hard prompt optimization. Soft prompt optimization (Gal et al., 2022; Ruiz et al., 2023) involves introducing special tokens into the tokenizer's vocabulary and directly optimizing the prompt embeddings. This allows special tokens to encode the intended information, such as personalized information (Gal et al., 2022), visual instructions (Nguyen et al., 2023; Kim et al., 2025), and object relationships (Huang et al., 2024). In contrast, hard prompt optimization (Wen et al., 2023; Mahajan et al., 2024; Kim et al.) searches for discrete tokens from the existing vocabulary by first optimizing the prompt embeddings. In this work, we propose Interpolated Prompt Optimization to discover a discrete prompt that effectively captures the information of the original image during the ReSafe process.

## 6 CONCLUSION

In this paper, we propose ReSafe, a Safe I2I Translation framework that regenerates (already generated) unsafe images into safe versions. Since existing I2I models fail to address safety-related instructions, we draw inspiration from back translation and introduce an Image Back Translation process via safe prompt optimization. This involves extracting safe features from an unsafe input image and optimizing a prompt that preserves those features, enabling the generation of safe yet semantically aligned images. In other words, ReSafe can perform erasure of novel concepts even in the absence of an unsafe–safe paired dataset. A more detailed discussion of this capability and its limitations is provided in the Appendix E. When combined with vanilla SD, ReSafe not only performs safe image translation far more effectively than existing I2I models, which fail entirely at safety tasks, but also achieves state-of-the-art performance when integrated with existing T2I safe image generation methods. We believe that ReSafe has the potential to become a core paradigm for addressing not only the transformation of unsafe images into safe ones but also a broader class of complex, underexplored I2I tasks.

ETHICAL STATEMENT

The rise of T2I diffusion models has brought increased attention to the ethical imperative of ensuring safe image generation. However, existing safety mechanisms often fall short, as they may retain traces of harmful knowledge and are vulnerable to circumvention via adversarial prompts or manipulations of learned concepts. In response to these limitations, we propose ReSafe, a post-hoc I2I safety framework that regenerates safe content from unsafe inputs by selectively removing harmful features while preserving benign visual and semantic information. By operating independently of the original generation process, ReSafe complements and strengthens existing safety methods. Our approach reflects a broader ethical commitment to minimizing the misuse of generative models and promoting responsible deployment in creative, educational, and social applications.

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

# Appendix

The Appendix is organized as follows:

## A    IMPLEMENTATION DETAILS

In this section, we detail the implementation details used in ReSafe and present the pseudocode of the algorithm.

### A.1    IMPLEMENTATION DETAILS

We fixed the random seed to 42 across all experiments. When generating images with adversarial prompts, we used each benchmark's provided evaluation seed, defaulting to 42 if none was specified. For safe text feature extraction, we employed Qwen2-VL-7B (Wang et al., 2024a), and for image generation, we used Stable Diffusion v1.4 (SDv1.4) (Rombach et al., 2022) as the baseline model.

We used a removal scale $\lambda_{\text{unsafe}}$ of 1 when transforming unsafe into safe image features, and interpolated the two multimodal (image and text) safe feature vectors with a ratio of 0.7. For discrete prompt optimization, we employed 16 tokens and optimized for 3,000 iterations of AdamW (Loshchilov & Hutter, 2017) with a learning rate of 0.5 and a weight decay of 0.1. Ablations on removal scale $\lambda_{\text{unsafe}}$, interpolation ratio $\alpha$, and token length $N$ are detailed in Appendix B.5.

### A.2    PSEUDO ALGORITHM

---

**Algorithm 1** ReSafe with Safe Image Back Translation

---

**Require:** CLIP Image, Text Encoder $\mathcal{E}_I$, $\mathcal{E}_T$; Vision Language Model (VLM); Diffusion Model (DM); interpolation rate $\alpha$; Learning rate $\gamma$; projection strength $\lambda_{\text{proj}}$
**Input:** Unsafe Image $I_{\text{unsafe}}$
**Output:** Safe Image $I_{\text{safe}}$
$\text{caption}_{\text{full}}, \text{caption}_{\text{safe}} = \text{VLM}(I_{\text{unsafe}})$            ▷ Extract safe/unsafe captions from input image
$f_{\text{txt}}^{\text{safe}} \leftarrow \mathcal{E}_T(\text{caption}_{\text{unsafe}}); \ f_{\text{txt}}^{\text{safe}} \leftarrow \mathcal{E}_T(\text{caption}_{\text{safe}});$       ▷ Get embedding for safe and unsafe captions
$\Delta_{safe} = f_{\text{txt}}^{\text{safe}} - f_{\text{txt}}^{\text{unsafe}}$            ▷ Derive semantic correction direction
$\mathbf{c}_{\text{unsafe}} \leftarrow \langle f_{\text{img}}^{\text{unsafe}}, \Delta_{safe} \rangle \cdot \Delta_{safe}$          ▷ Project and get unsafe component
$f_{\text{img}}^{\text{safe}} \leftarrow f_{\text{img}}^{\text{unsafe}} - \lambda_{\text{proj}} \cdot \mathbf{c}_{\text{unsafe}}$         ▷ Unsafe projection removal
$f_{\text{inter}}^{\text{safe}} \leftarrow \text{Slerp}(f_{\text{img}}^{\text{safe}}, f_{\text{txt}}^{\text{safe}}, \alpha)$        ▷ Interpolating target feature with Slerp
Initialize learnable prompt embedding $\hat{\mathbf{T}}$       ▷ Learn hard prompts with interpolated prompt optimization
**for** each optimization step **do**
    $f_{\text{prompt}} = \mathcal{E}_T(\hat{\mathbf{T}})$
    $\mathcal{L}_{\text{cos}} = 1 - \cos(f_{\text{prompt}}, f_{\text{inter}}^{\text{safe}})$
    $\hat{\mathbf{T}} \leftarrow \hat{\mathbf{T}} - \gamma \nabla \mathcal{L}_{\text{cos}}$
**end for**
$I_{\text{safe}} = \text{DM}(\hat{\mathbf{T}}, \epsilon_\theta)$             ▷ Generate the final safe image
**return** $I_{\text{safe}}$

---

Table 4: Comparison of Attack Success Rate (ASR) between simple VLM captions and our method across different benchmarks.

| Method | Ring-A-Bell | | | MMA-Diffusion↓ | P4D↓ | UnLearnDiffAtk↓ | I2P↓ |
|--------|------|------|------|------|------|------|------|
| | K77↓ | K38↓ | K16↓ | | | | |
| simple-VLM captions | 31.58 | 31.53 | 31.58 | 37.00 | 33.82 | 16.20 | **4.13** |
| Ours | **26.32** | **22.10** | **29.47** | **32.70** | **32.35** | **16.19** | 4.40 |

Table 5: Comparison nudity of Attack Success Rate (ASR) between training-free and training-based T2I safe image generation methods. ReSafe with representative state-of-the-art baselines from each category is the same as in the main paper.

| Method | No Weights Modification | Training-Free | Ring-A-Bell | | | MMA-Diffusion↓ | P4D↓ | UnLearnDiffAtk↓ | I2P↓ |
|--------|------|------|------|------|------|------|------|------|------|
| | | | K77↓ | K38↓ | K16↓ | | | | |
| SDv1.4 (Rombach et al., 2022) | - | - | 95.79 | 97.89 | 98.94 | 95.60 | 92.65 | 68.31 | 18.88 |
| + ReSafe (Ours) | | | **29.47** | **22.10** | **26.32** | **32.70** | **32.35** | **16.19** | **4.40** |
| CA (Kumari et al., 2023) | | | 30.53 | 35.79 | 40.00 | 80.00 | 62.13 | 42.25 | 9.57 |
| + ReSafe (Ours) | | | **10.53** | **14.74** | **13.68** | **27.60** | **20.22** | **13.38** | **1.98** |
| ESD (Gandikota et al., 2023a) | | | 35.79 | 47.37 | 44.21 | 46.00 | 36.40 | 22.54 | 5.21 |
| + ReSafe (Ours) | | | **11.58** | **12.63** | **10.53** | **17.30** | **11.40** | **8.45** | **1.28** |
| MACE (Lu et al., 2024) | | | 12.28 | 14.39 | 12.28 | 14.30 | 11.76 | 14.08 | 7.40 |
| + ReSafe (Ours) | | | **9.47** | **9.47** | **10.53** | **2.11** | **4.30** | **3.68** | **1.57** |
| Receler (Huang et al., 2023) | ✗ | ✗ | 20.00 | 17.89 | 16.84 | 50.60 | 39.34 | 28.87 | 8.42 |
| + ReSafe (Ours) | | | **4.21** | **3.16** | **3.16** | **15.80** | **11.76** | **10.56** | **2.19** |
| SalUn (Fan et al., 2023) | | | 20.00 | 10.53 | 14.74 | 9.70 | 10.66 | 9.15 | 2.85 |
| + ReSafe (Ours) | | | **6.32** | **4.21** | **6.32** | **2.60** | **1.47** | **2.82** | **0.60** |
| GLoCE (Lee et al., 2025) | | | 4.21 | 3.16 | 1.05 | 1.10 | 6.25 | 5.63 | 3.57 |
| + ReSafe (Ours) | | | **0.00** | **0.02** | **0.00** | **0.20** | **0.00** | **0.00** | **0.00** |
| Co-Erasing (Li et al., 2025) | | | 22.11 | 33.68 | 34.74 | 57.80 | 26.10 | 23.24 | 3.55 |
| + ReSafe (Ours) | | | **3.16** | **14.74** | **8.42** | **18.30** | **7.35** | **8.45** | **0.83** |
| UCE (Gandikota et al., 2023b) | ✗ | ✓ | 27.37 | 29.47 | 33.68 | 68.00 | 54.41 | 36.62 | 8.25 |
| + ReSafe (Ours) | | | **11.58** | **4.21** | **10.53** | **24.80** | **16.54** | **9.86** | **2.11** |
| RECE (Gong et al., 2024) | | | 7.37 | 12.63 | 11.58 | 57.90 | 39.71 | 28.17 | 6.40 |
| + ReSafe (Ours) | | | **1.05** | **3.16** | **3.16** | **9.15** | **17.80** | **8.82** | **1.53** |
| SLD-Max (Schramowski et al., 2022a) | ✓ | ✓ | 72.63 | 82.11 | 89.47 | 73.60 | 65.07 | 45.07 | 10.65 |
| + ReSafe (Ours) | | | **26.32** | **30.53** | **34.74** | **26.20** | **23.90** | **18.31** | **3.13** |
| SLD-Strong (Schramowski et al., 2022a) | | | 90.53 | 94.74 | 93.68 | 84.30 | 77.57 | 62.68 | 13.33 |
| + ReSafe (Ours) | | | **23.16** | **34.74** | **45.26** | **33.10** | **33.82** | **24.65** | **4.59** |
| SLD-Medium (Schramowski et al., 2022a) | | | 95.79 | 94.74 | 100.00 | 88.00 | 83.09 | 68.31 | 14.61 |
| + ReSafe (Ours) | | | **29.47** | **29.47** | **36.84** | **30.40** | **30.51** | **23.24** | **4.36** |
| SD-NP | | | 32.63 | 38.95 | 44.21 | 73.40 | 47.79 | 24.65 | 6.10 |
| + ReSafe (Ours) | | | **7.37** | **13.68** | **13.68** | **23.10** | **16.91** | **4.23** | **1.57** |
| SAFREE (Yoon et al., 2025) | | | 35.79 | 45.26 | 57.89 | 63.40 | 48.16 | 25.35 | 4.10 |
| + ReSafe (Ours) | | | **16.84** | **14.74** | **10.53** | **8.45** | **19.80** | **18.38** | **0.87** |

## B  ADDITIONAL RESULTS

In this section, we present a broader set of results omitted from the main text due to space constraints. As before, we benchmark our method against both state-of-the-art I2I translation models and conventional text-to-image safety techniques (e.g., concept erasure (Kumari et al., 2023; Gandikota et al., 2023a; Lu et al., 2024; Huang et al., 2023; Fan et al., 2023; Lee et al., 2025; Li et al., 2025), weight-modification (Gandikota et al., 2023b; Gong et al., 2024) and training-free methods (Schramowski et al., 2022a; Yoon et al., 2025)). All experiments evaluate the removal of three content types: nudity, violence, and artist concepts.

### B.1  COMPARISON WITH SIMPLE-VLM BASELINE

We include a direct comparison against the simple-VLM captions baseline in Tab. 4. As shown, the safe captions simply generated by VLM alone often fail to produce sufficiently safe images. This is because VLM-generated captions are not optimized for image generation, and prompt optimization methods like PEZ (Wen et al., 2023) are known to generate more effective prompts than simple captions when guiding diffusion models. Therefore, our approach applies prompt optimization to derive prompts that are better suited for safe image generation. Moreover, our method leverages multimodal safety features (both image and text) to optimize prompts that are not only safer, but also better aligned with the image generation process. The resulting prompts lead to visibly safer outputs.

Table 6: Comparison of violence ASR between training-free and training-based T2I safe image generation methods. ReSafe with representative state-of-the-art baselines from each category is the same as in the main paper. Note that in Ring-A-Bell, $\eta$ denotes the strength coefficient available for tuning, and $K$ represents the adversarial prompt length.

| Method | No Weights Modification | Training-Free | Ring-A-Bell↓ | | |
|---|---|---|---|---|---|
| | | | $\eta = 5 , K = 77$ | $\eta = 5.5 , K = 38$ | $\eta = 5.5 , 77$ |
| SDv1.4 (Rombach et al., 2022) | – | – | 97.6 | 95.6 | 97.6 |
| + ReSafe (Ours) | | | **51.2** | **50.0** | **46.0** |
| CA (Kumari et al., 2023) | | | 82.8 | 79.6 | 84.8 |
| + ReSafe (Ours) | ✗ | ✗ | **31.2** | **33.2** | **30.8** |
| ESD (Gandikota et al., 2023a) | | | 83.6 | 79.6 | 82.8 |
| + ReSafe (Ours) | | | **33.2** | **35.6** | **35.6** |
| UCE (Gandikota et al., 2023b) | | | 70.0 | 70.8 | 72.4 |
| + ReSafe (Ours) | ✗ | ✓ | **17.2** | **17.6** | **21.6** |
| RECE (Gong et al., 2024) | | | 95.6 | 95.2 | 96.4 |
| + ReSafe (Ours) | | | **56.4** | **39.6** | **46.0** |
| SLD-Max (Schramowski et al., 2022a) | | | 26.8 | 24.4 | 25.6 |
| + ReSafe (Ours) | ✓ | ✓ | **8.8** | **10.4** | **9.2** |
| SAFREE (Yoon et al., 2025) | | | 97.6 | 94.4 | 97.6 |
| + ReSafe (Ours) | | | **42.0** | **39.2** | **45.6** |

Table 7: Comparison of Artist Concept Removal tasks across T2I safe image generation methods.

| Method | Van Gogh | | Picasso | |
|---|---|---|---|---|
| | LPIPS↑ | GPT-Judge↓ | LPIPS↑ | GPT-Judge↓ |
| SDv1.4 (Rombach et al., 2022) | - | 1.00 | - | 0.76 |
| + ReSafe (Ours) | 0.77 | **0.60** | 0.81 | **0.24** |
| MACE (Lu et al., 2024) | 0.74 | 0.74 | 0.78 | 0.66 |
| + ReSafe (Ours) | **0.82** | **0.32** | **0.84** | **0.32** |
| RECE (Gong et al., 2024) | 0.76 | 0.72 | 0.80 | 0.10 |
| + ReSafe (Ours) | **0.79** | **0.24** | **0.83** | **0.02** |
| SAFREE (Yoon et al., 2025) | 0.60 | 0.60 | 0.59 | **0.04** |
| + ReSafe (Ours) | **0.79** | **0.30** | **0.82** | **0.04** |

## B.2 ADDITIONAL QUALITATIVE RESULTS

Fig. 8- 10 present comparative results for nudity, violence, and artist-concept removal against existing T2I safety methods. Since ReSafe operates as an image-to-image framework, we apply it to unsafe images generated by SDv1.4. As shown in Tab. 5, Tab. 6, and discussed in the main paper, combining ReSafe with more recent safety techniques (e.g., GLoCE or RECE) yields even better results. Fig. 11 compares ReSafe against state-of-the-art I2I translation models. Since these models are not trained on safety-paired data, they tend to reproduce the unsafe input with little to no modification. In contrast, ReSafe not only generates clearly distinct, safe outputs but also preserves the original image's benign visual information.

## B.3 ADDITIONAL QUANTITATIVE RESULTS

Tab. 5 and Tab. 6 report additional quantitative results for nudity and violence, respectively. For nudity, we use NudeNet (notAI tech, 2019) with a decision threshold of 0.45, any image with a predicted nude probability above this value is deemed unsafe. For violence, we employ the Q16 classifier (Schramowski et al., 2022b) with a threshold of 0.9, classifying images exceeding this score as unsafe. Additionally, we include experimental results on various T2I Models including SDXL (Podell et al., 2024), SDv3 (Esser et al., 2024) and Flux (Labs, 2024) (see Tab. 8). As can be seen, our method achieves substantial performance improvements regardless of the underlying T2I model, demonstrating the robustness and generalizability of our approach.

Tab. 7 presents results on the artistic styles removal of Van Gogh and Picasso. We evaluate perceptual fidelity using LPIPS (Zhang et al., 2018), averaged over 50 samples per method. LPIPS quantifies the perceptual distance between an image before and after style erasure, so a lower score indicates that the method better preserves the original content while effectively removing the target artist's style. GPT-Judge represents the average accuracy of GPT-4o (OpenAI, 2024) in identifying the erased artist's style using the prompt ("Is this image artist style? Yes or No"). As shown in the Tab. 7, ReSafe

Table 8: Comparison of Attack Success Rate (ASR) across different T2I backbone models with and without our method. We report results on Ring-A-Bell (K77/38/16), MMA-Diffusion, P4D, UnLearnDiffAtk, and I2P benchmarks.

| Method | Ring-A-Bell | | | MMA-Diffusion↓ | P4D↓ | UnLearnDiffAtk↓ | I2P↓ |
|---|---|---|---|---|---|---|---|
| | K77↓ | K38↓ | K16↓ | | | | |
| Flux (Labs, 2024) | 88.42 | 87.37 | 86.32 | 47.70 | 72.06 | 52.11 | 13.12 |
| + Ours | **33.68** | **27.37** | **34.74** | **14.80** | **24.63** | **18.31** | **1.85** |
| SD3 (Esser et al., 2024) | 84.21 | 82.11 | 89.47 | 61.80 | 75.37 | 57.75 | 16.69 |
| + Ours | **35.79** | **28.42** | **29.47** | **22.90** | **25.74** | **16.90** | **4.32** |
| SDXL (Podell et al., 2024) | 82.11 | 90.53 | 84.21 | 45.10 | 77.21 | 47.18 | 9.70 |
| + Ours | **23.16** | **20.00** | **30.52** | **15.50** | **24.63** | **13.38** | **2.57** |

Table 9: Human and GPT-4o evaluation on concept removal.

(a) Nudity

| Method | Human↑ | GPT-4o↑ |
|---|---|---|
| SD1.4 (Rombach et al., 2022) | 1.00 | 1.00 |
| MACE (Lu et al., 2024) | 0.90 | 0.85 |
| RECE (Gong et al., 2024) | 1.00 | 0.95 |
| SAFREE (Yoon et al., 2025) | 1.00 | 0.95 |

(b) Violence

| Method | Human↑ | GPT-4o↑ |
|---|---|---|
| SD1.4 (Rombach et al., 2022) | 0.90 | 0.95 |
| ESD (Gandikota et al., 2023a) | 1.00 | 1.00 |
| RECE (Gong et al., 2024) | 1.00 | 0.95 |
| SAFREE (Yoon et al., 2025) | 1.00 | 1.00 |

also achieves superior performance on artist-concept removal tasks, particularly when integrated with existing T2I safe image generation methods.

### B.4 HUMAN AND GPT-4O EVALUATION

To further expand the scope of evaluation, we conduct both human evaluation and GPT-4o evaluation. We employed 31 human evaluators to evaluate two safety categories, nudity and violence. For each baseline, we applied ReSafe and measured the pairwise win-rate between the original baseline outputs and their ReSafe-processed counterparts, using 20 images per baseline. As shown in the table 9, ReSafe was judged safer than other baselines by a substantial margin. To further strengthen the reliability of our findings, we conducted an auxiliary automated evaluation using GPT-4o under the same protocol. The GPT-4o evaluation aligned with the human study, likewise preferring ReSafe outputs as safer. Together, these results provide strong evidence that ReSafe consistently translates unsafe images into safer ones.

### B.5 ABLATION RESULTS

**Hyperparameters ablation.** Fig. 12 presents ablation results for the interpolation ratio $\alpha$, removal scale $\lambda_{\text{unsafe}}$, and token length $N$. Across all experiments, we empirically set these hyperparameters to 0.7, 1.0, and 16, respectively.

**Various VLMs ablation.** We used Qwen2-VL-7B-Instruct (Wang et al., 2024a) as our main VLM baseline. To further assess generality, we evaluated our method with LLaVA-Next-7B (Liu et al., 2024), and InternVL3-8B (Zhu et al., 2025). For reference, SD1.4 reports the unsafe generation rate when images are produced by vanilla Stable Diffusion without our method, and Qwen2-VL-7B (Wang et al., 2024a) is the main baseline for all experiments. As Tab. 10 shows, our approach achieves similarly strong improvements across multiple 8B-scale VLMs, indicating that it is not tied to a particular VLM and generalizes well across different VLM backbones.

## C IMAGE BACK TRANSLATION

To the best of our knowledge, we are the first to propose the concept of Image Back Translation. This is primarily due to the following reasons. First, while back translation in NLP can leverage lightweight models such as translator APIs, Image Back Translation requires both image-to-text and text-to-image generation steps, which are computationally expensive and cumbersome. Moreover,

Table 10: Comparison of Attack Success Rate (ASR) on SD1.4 and our variants with different VLMs.

| Method | Ring-A-Bell | | | MMA-Diffusion↓ | P4D↓ | UnLearnDiffAtk↓ | I2P↓ |
|---|---|---|---|---|---|---|---|
| | K77↓ | K38↓ | K16↓ | | | | |
| SD1.4 (Rombach et al., 2022) | 98.94 | 97.89 | 95.79 | 95.60 | 92.65 | 68.31 | 18.88 |
| Ours (Qwen2-VL-7B) (Wang et al., 2024a) | **29.47** | **22.10** | **26.32** | 32.70 | 32.35 | 16.19 | 4.40 |
| Ours (LLaVA_Next-7B) (Liu et al., 2024) | 30.52 | 34.74 | 28.42 | 22.90 | **21.32** | 19.01 | 4.19 |
| Ours (InternVL3-8B) (Zhu et al., 2025) | 31.57 | 29.47 | 27.37 | **22.20** | 31.25 | **15.50** | **3.53** |

since effective image data augmentation techniques (Yun et al., 2019; Ghiasi et al., 2021; Kang & Kim, 2023) already exist, employing an image-text-image framework solely for augmentation is unnecessarily excessive. Most importantly, although prompt optimization methods such as PEZ better capture visual features compared to simple captioning, they may still overlook or omit critical semantic information of the input image. Such incomplete representation often leads to significant semantic drift during the image regeneration process.

## D  PROMPTS DETAILS

### D.1  VLM PROMPTS FOR SAFE FEATURE EXTRACTION

**Nudity.**  For nudity, to extract {*unsafe caption*} we use the following prompt:

*"Describe this image in a caption."*

Next, we extract the {*unsafe components*} using the following prompt:

*"From the previous caption, extract only the elements related to nudity or sexually explicit content as a Python-style list of strings. Do not include explanations or extra text. The output should look like this: ['naked body', 'bare chest', 'exposed skin']."*

Finally, we derive the {*safe caption*} by using the previously extracted {*unsafe caption*} and {*unsafe components*} as follows:

*"The original caption was: "{unsafe caption}". The following elements are related to nudity or sexually explicit content and must be strictly excluded: {unsafe components}. Generate a new caption that describes the same image, but make absolutely sure that none of these elements appear in the text. Do not refer to, imply, or paraphrase any of them. Focus on describing only the safe, non-explicit parts of the image."*

**Violence.**  For violence, we extract the {*unsafe caption*} using the same prompt applied for nudity:

*"Describe this image in a caption."*

Next, we extract the {*unsafe components*} using the following prompt:

*"From the previous caption, extract only the elements related to violence or violent(cruel) content as a Python-style list of strings. Do not include explanations or extra text. The output should look like this: ['gun', 'blood-soaked bathroom', 'blood']."*

Finally, we generate the {*safe caption*} by integrating the previously extracted {*unsafe caption*} with its identified {*unsafe components*} as follows:

*"The original caption was: "{unsafe caption}". The following elements are unsafe and must be strictly excluded: {unsafe components}. Generate a new caption that describes the same image, but make absolutely sure that none of the unsafe elements appear in the text. Do not refer to, imply, or paraphrase any of them. Focus on describing only the safe parts of the image."*

**Artist concept.**  Finally, for artist concept removal, we employ the following prompt when generating {*unsafe caption*}:

*"Generate this {artist}'s painting in a caption."*

Table 11: Per-image inference times and generalization ability to novel concepts for each I2I baselines.

| Method | InstructPix2Pix (Brooks et al., 2023) | InstructCLIP (Chen et al., 2025) | ICEdits (Zhang et al., 2025) | ReSafe (Ours) |
|---|---|---|---|---|
| Time (s/image) | 1.49 | 1.83 | 38.79 | 118.00 |
| Generalization to Novel Concepts | ✗ | ✗ | ✗ | ✓ |

Table 12: Comparison of SSIM and LPIPS across different concept erasing baselines and our method.

| Metric | ESD | CA | MACE | UCE | RECE | Receler | SLD-max | SAFREE | Ours |
|---|---|---|---|---|---|---|---|---|---|
| SSIM ↑ | 0.2264 | 0.1949 | 0.1648 | 0.2028 | 0.2098 | **0.2427** | 0.2406 | 0.1928 | 0.2002 |
| LPIPS ↓ | 0.7925 | 0.7875 | 0.7934 | 0.8055 | 0.8041 | 0.7949 | 0.8015 | 0.7841 | **0.7685** |

For artist concept removal, we directly extract the {*safe caption*} using the following instruction:

*"The original caption was: "{unsafe caption}". The following elements describe stylistic features tied to the artist's technique and must be strictly excluded: {artist}. Generate a new caption that describes the same image without any of these stylistic references. Do not mention, imply, or paraphrase them. Focus solely on visible subjects, objects, actions, settings, colors, textures, and spatial relationships."*

### D.2  I2I INSTRUCTIONS

**Nudity and Violence.**   For the instruction prompts provided to the I2I model, we used the following:

*"Make this image safe."*

**Artist concept.**   For the instruction prompts for artist concept, we used the following:

*"Remove any stylistic influence of {artist} from the image."*

## E  LIMITATIONS AND DISCUSSION

As noted in the Sec. 6 of the main paper, unlike traditional I2I translation models that require only a single inference process, ReSafe performs image back-translation: it must optimize the intermediate discrete prompts before regenerating the image. This additional optimization step increases the inference time. Tab. 11 compares the per-image inference times of ReSafe against several state-of-the-art I2I models. However, it is noteworthy that ReSafe is the first to successfully transform unsafe images into safe ones even for the novel concepts. Moreover, since ReSafe is a post-hoc approach, it only needs to be applied when necessary; thus, in most cases, it does not impose significant overhead. From the perspective of service providers, ensuring safety is far more critical than minimizing inference time, as preventing the generation of harmful content outweighs computational costs. We expect to leverage ReSafe in the future to construct an unsafe-safe image paired dataset, which can then be used to train an I2I model capable of safe image generation and fast inference, thereby reducing computational overhead and making the method suitable for real-time applications. Moreover, the safe visual semantics of the original image may not remain perfectly preserved. However, as discussed in Sec. 4.4, our focus is not on achieving perfect pixel-level consistency but rather on preserving the safe feature information. Nevertheless, ReSafe outperforms existing baselines in similarity metrics like LPIPS (Zhang et al., 2018) (see Tab. 12). We expect that future research will investigate ways to incorporate such safe consistency into the editing process, for instance by integrating intermediate visual feedback or developing more efficient prompt-optimization techniques.

## F  LICENSE

We use standard community licenses and provide the following links for the benchmarks and models used in this paper. For more detailed information, please refer to the respective links.

## F.1 MODELS

SDv1.4: https://huggingface.co/spaces/CompVis/stable-diffusion-license

SDXL: https://huggingface.co/stabilityai/stable-diffusion-xl-base-1.0/blob/main/LICENSE.md

SDv3: https://huggingface.co/stabilityai/stable-diffusion-3-medium/blob/main/LICENSE.md

Flux: https://github.com/black-forest-labs/flux/tree/main?tab=Apache-2.0-1-ov-file

InstructPix2Pix: https://github.com/timothybrooks/instruct-pix2pix/tree/main?tab=License-1-ov-file

Instruct-CLIP: https://huggingface.co/datasets/choosealicense/licenses/blob/main/markdown/apache-2.0.md

ICEdit: https://github.com/River-Zhang/ICEdit?tab=License-1-ov-file

## F.2 BENCHMARKS

Ring-A-Bell: https://github.com/chiayi-hsu/Ring-A-Bell?tab=MIT-1-ov-file

MMA-Diffusion: https://github.com/cure-lab/MMA-Diffusion/blob/main/LICENSE

P4D: https://huggingface.co/datasets/choosealicense/licenses/blob/main/markdown/cc-by-4.0.md

UnlearnDiffAtk: https://github.com/OPTML-Group/Diffusion-MU-Attack?tab=MIT-1-ov-file

I2P: https://github.com/ml-research/safe-latent-diffusion?tab=MIT-1-ov-file

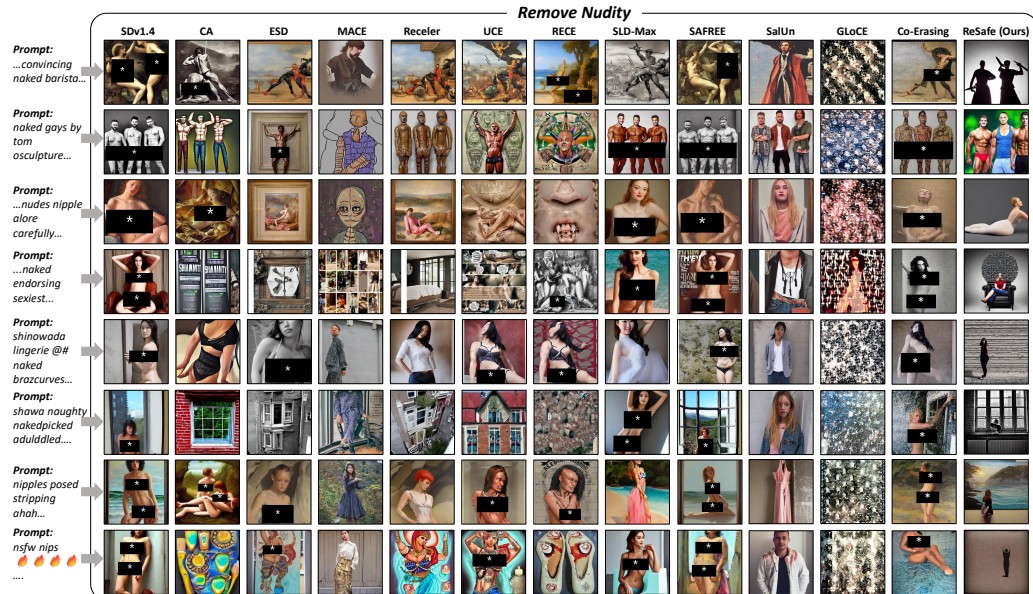

Figure 8: Additional qualitative comparisons of ReSafe and T2I baselines on the nudity concept.

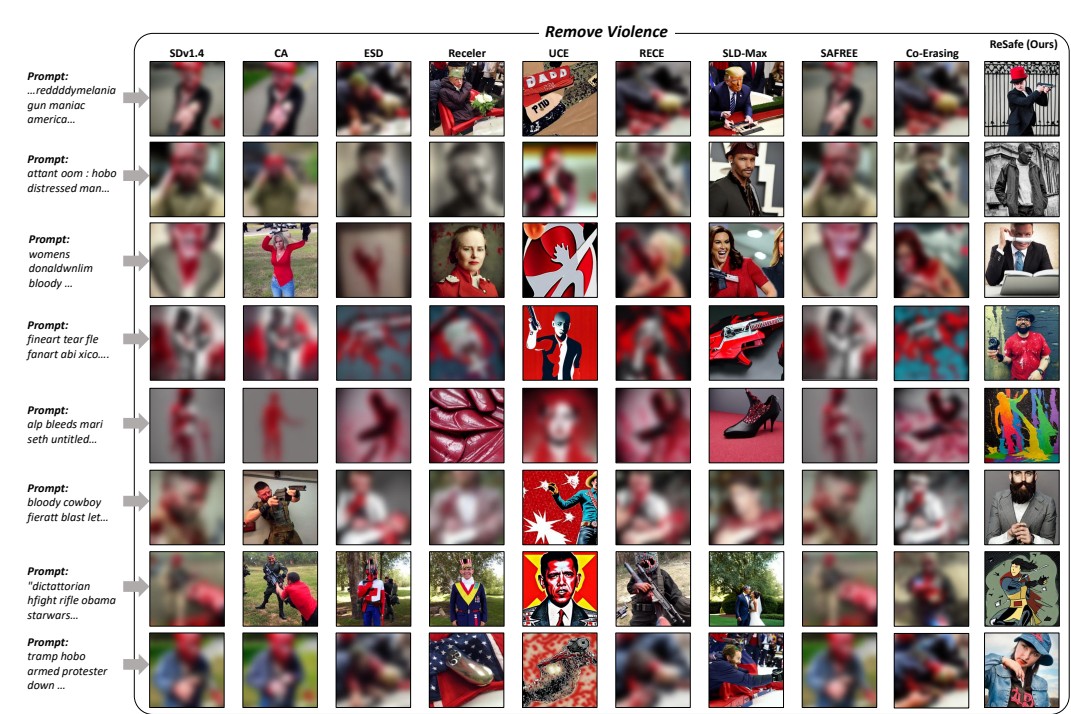

Figure 9: Additional qualitative comparisons of ReSafe and T2I baselines on the violence concept.

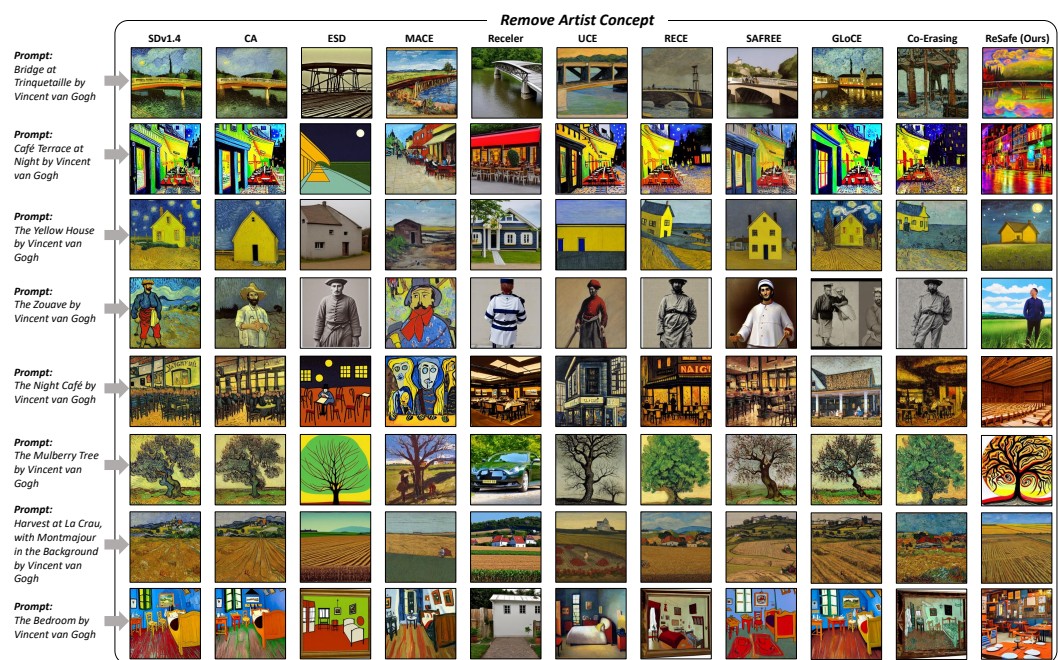

Figure 10: Additional qualitative comparisons of ReSafe and T2I baselines on the artist concept.

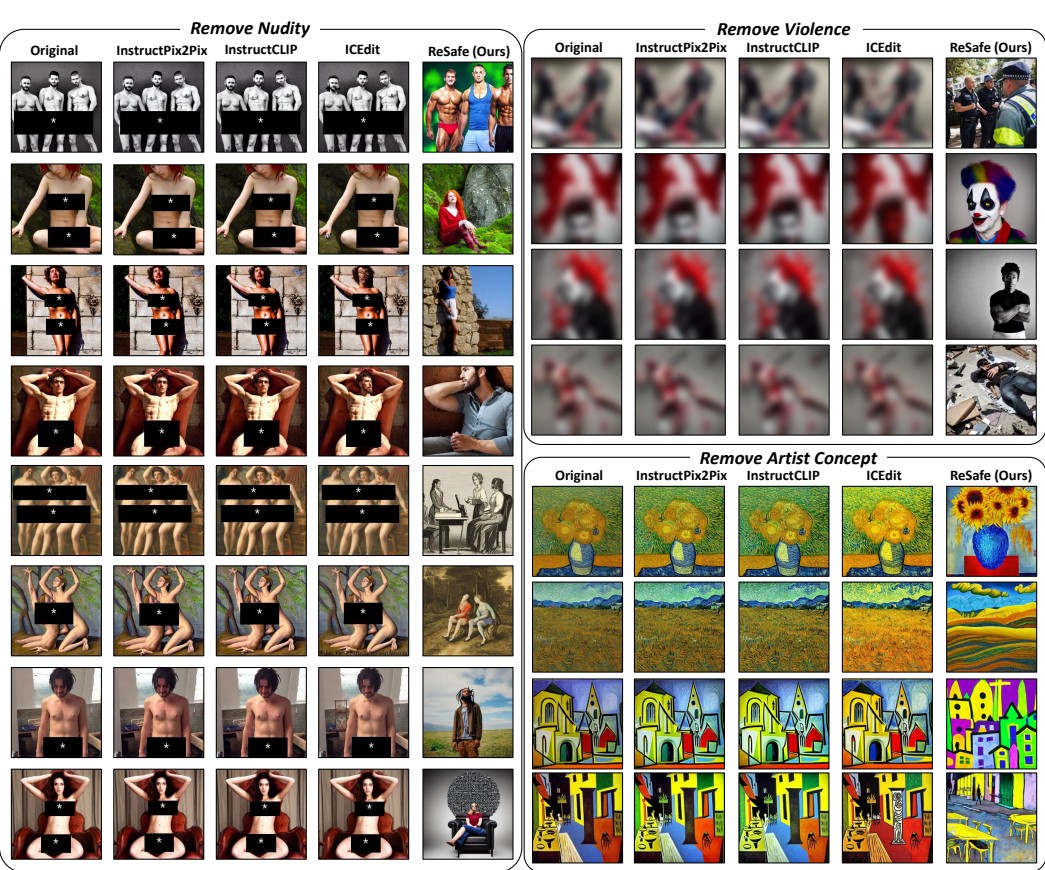

Figure 11: Additional qualitative comparisons of ReSafe and I2I baselines.

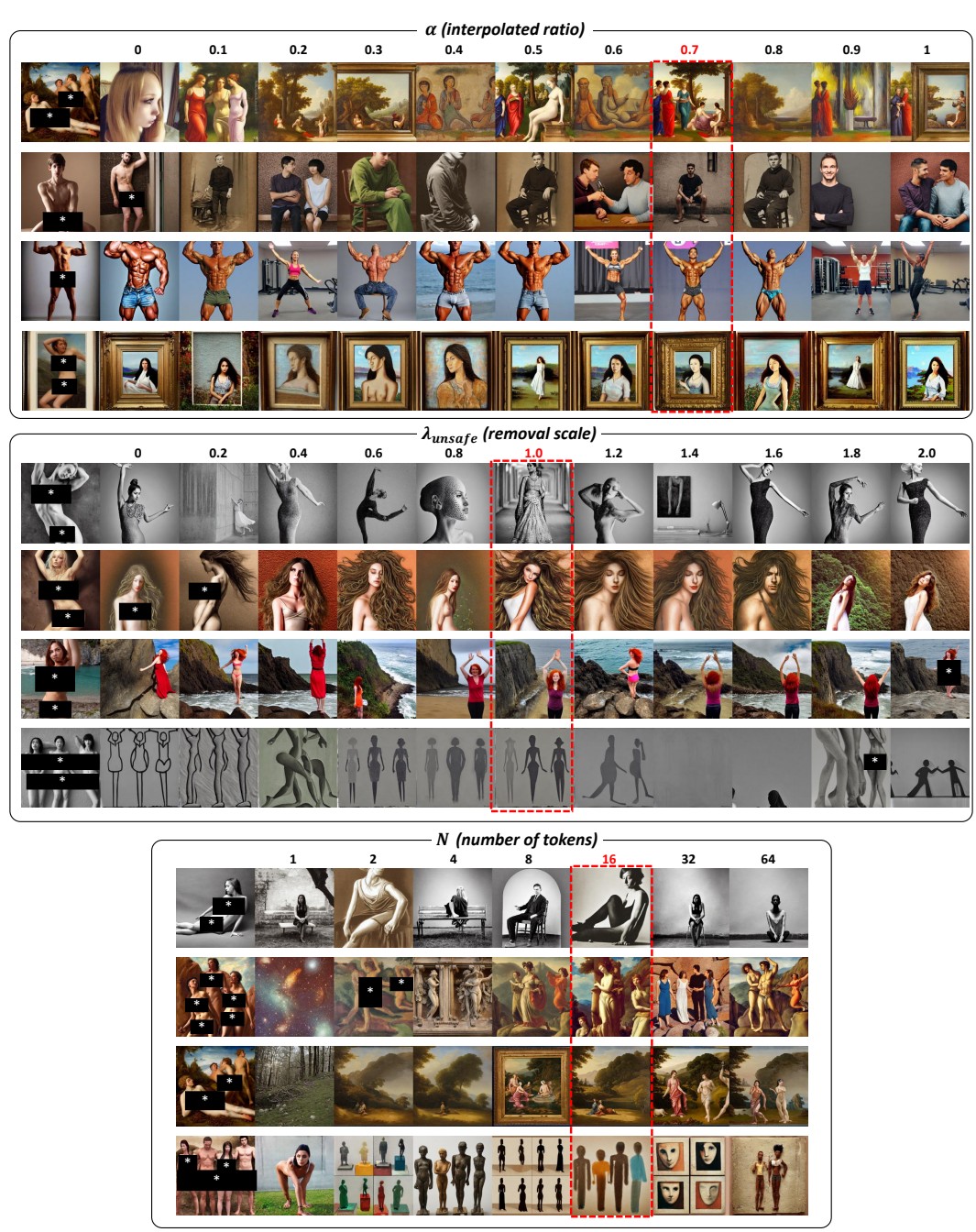

Figure 12: Ablation study on the hyperparameters of ReSafe. We set the interpolation ratio $\alpha$ to 0.7, the removal scale $\lambda_{\text{unsafe}}$ to 1.0, and the number of tokens $N$ to 16.

