# OpenReview forum: "ReSafe: Enhancing Safety of Text-to-Image Diffusion via Post-Hoc Image Back Translation"
_ICLR.cc/2026/Conference — ICLR 2026 Conference Withdrawn Submission_

### Official Review · Reviewer_8FAe · 2025-10-27

**Soundness:** 3
**Presentation:** 3
**Contribution:** 3
**Rating:** 6
**Confidence:** 4

**Summary:**

The paper introduces ReSafe, a post-hoc Image-to-Image (I2I) safety framework designed to convert unsafe images generated by text-to-image (T2I) diffusion models into safe ones. Unlike existing safety methods that attempt to prevent unsafe content during T2I generation, ReSafe repairs unsafe outputs. ReSafe achieves 3–7× lower Attack Success Rates (ASR) across adversarial benchmarks. The authors claim it selectively removes unsafe content while preserving safe semantics without modifying model weights.

**Strengths:**

1. Technical novelty: The combination of CLIP-based unsafe feature projection, multimodal (text+image) interpolation, and discrete prompt optimization is a fresh and coherent integration of known techniques.

2. Model-agnostic and modular: Can be applied after any diffusion model without retraining or weight modifications, making it versatile.

3. Strong empirical results: Shows consistent 3–7× reduction in ASR across multiple challenging adversarial benchmarks.

**Weaknesses:**

1. Justification gap: The need for I2I post-processing is not strongly defended against the alternative of improving T2I prompt filtering or adversarial robustness directly.

2. Heavy pipeline complexity: Multiple stages (VLM captioning, projection, prompt optimization, second T2I pass) may hurt latency and practicality; no cost analysis provided.

3. **Dependence on caption reliability: If the VLM fails to identify unsafe concepts, subsequent steps may be ineffective.**

4. Sensitivity to hyperparameters: The approach involves several important hyperparameters. While some qualitative insights are provided in the appendix, a more systematic quantitative analysis of how these parameters influence ASR and image quality would help improve transparency and robustness.

5. Runtime and practical trade-offs: Since the pipeline involves multiple stages (e.g., VLM captioning, feature projection, prompt optimization, and re-generation via T2I), the overall runtime may be significantly higher than single-pass safety methods. If the method incurs noticeable latency, it would be helpful for the main paper to discuss this as a trade-off and possibly provide runtime comparisons or complexity estimates.

**Questions:**

1. Under what deployment scenarios is ReSafe more suitable than improved T2I defenses?

2. Dependence on safe caption accuracy: The method relies on a VLM to differentiate between unsafe and safe content in captions. How robust is the approach when the VLM fails to identify subtle or implicit unsafe concepts, or is itself subjected to adversarial prompt attacks? If available, could the authors provide quantitative results or a sensitivity analysis under such conditions?

3. Effects of hyper-parameter choices (Please refer weakness)

4. The approach assumes unsafe features are linearly separable via projection in CLIP embedding space. Would you examine potential entanglement between harmful and benign features that could lead to over-removal or semantic drift? Any visualizations or failure cases?

5. Clarification on semantic preservation beyond ASR: Tables 1 and 2 focus primarily on reducing Attack Success Rate (ASR), but this alone may not fully demonstrate that only unsafe content is removed while preserving the original semantics. Could the authors provide additional quantitative analysis of semantic or visual fidelity (e.g., CLIP/DINO similarity, latent feature overlap, or concept embedding proximity before and after I2I)?

6. Evaluating localized modification of unsafe regions: How do the authors assess whether their method selectively modifies only the unsafe regions of an image rather than causing broader semantic or visual alterations? If such an evaluation protocol exists (e.g., region-based analysis, concept localization, or attention/diffusion map comparison), could the authors report corresponding results for the experiments in Tables 1 and 2?

---

### Official Review · Reviewer_wno4 · 2025-10-28

**Soundness:** 2
**Presentation:** 3
**Contribution:** 2
**Rating:** 2
**Confidence:** 4

**Summary:**

This paper proposed ReSafe, a safety-oriented Image-to-Image translation framework for Text-to-Image (T2I) diffusion models. It addresses the limitations of existing T2I safety methods, such as their failure to fully eliminate harmful knowledge and vulnerability to adversarial prompts or concept arithmetic attacks, through a post-hoc image back-translation strategy. The framework first extracts multimodal safe features from unsafe inputs, then fuses these features via Spherical Linear Interpolation and optimizes discrete safe prompts. Ultimately, it generates images that remove harmful features while preserving safe semantic and visual information. The authors conducted experiments on five adversarial prompt benchmarks and demonstrated ReSafe’s effectiveness.

**Strengths:**

1. ReSafe is a post-hoc method that requires no modifications to underlying diffusion models and can be seamlessly integrated with existing T2I safety techniques. This plug-and-play property enables compatibility with mainstream diffusion models.
2. It supports the removal of multiple types of unsafe content, including nudity, violence, and specific artistic styles. Additionally, it handles both model-generated and externally provided unsafe images, covering more practical scenarios than T2I-only safety methods.

**Weaknesses:**

1. In my opinion, the motivation of this paper is illogical. If a T2I service provider detects that input prompts are unsafe or generated images are unsafe, they can refuse to output any images and inform the user that the service cannot be provided. I think this is sufficient.
2. ReSafe requires an additional discrete prompt optimization step, resulting in a per-image inference time that is significantly longer than that of I2I baselines. This limits its application in real-time or high-throughput scenarios.
3. Some formulas lack proper punctuation. For instance, Formulas (1), (2), and (3) are missing necessary punctuation. Please check the entire paper for such issues.

**Questions:**

1. ReSafe relies on unsafe content detectors. Only when the system can reliably detect that generated images are unsafe can ReSafe function properly. What is the authors’ perspective on scenarios where the system fails to stably detect unsafe content?

---

### Official Review · Reviewer_dRM6 · 2025-10-31

**Soundness:** 3
**Presentation:** 3
**Contribution:** 3
**Rating:** 6
**Confidence:** 3

**Summary:**

The paper introduces a post-hoc framework designed to regenerate safe images from unsafe ones generated by Text-to-Image (T2I) models.
Unlike prior methods that try to generate safe images from a text prompt, ReSafe takes an already-generated unsafe image as input and transforms it into a safe version by selectively removing only the harmful features while preserving the safe semantic and visual information.
Specifically, it uses a VLM to generate both unsafe and safe captions from the input image.
It then extracts safe visual and textual features by projecting out the unsafe direction from the image embedding, guided by the text embeddings.
To minimize information loss, it combines the safe image and text features. It then optimizes a discrete textual prompt to align with this interpolated safe feature vector. This optimized prompt effectively encodes the safe semantics of the original image.
As a post-hoc method, ReSafe is model-agnostic.

**Strengths:**

+ clear presentation
+ sound design
+ relatively comprehensive evaluation

**Weaknesses:**

1. The most significant practical weakness of ReSafe is its high computational cost and slow inference speed, which the authors explicitly acknowledge in the Appendix (Table 11). At 118 seconds per image, ReSafe is orders of magnitude slower than baseline I2I models like InstructPix2Pix (1.49s) or Instruct-CLIP (1.83s). This is a critical limitation for any real-world application where rapid content moderation is required.

2. The paper’s evaluation of how well ReSafe preserves safe semantic and visual information is insufficient. While it reports metrics like LPIPS and CLIP/DINOv2 similarity (Table 3), these are low-level perceptual metrics that do not adequately capture high-level semantic fidelity.

3. The entire ReSafe pipeline relies on the initial VLM correctly identifying and captioning unsafe elements. The paper does not investigate whether this front-end is robust to adversarial attacks. An adversary could potentially create an image that is visually unsafe to humans but is mis-captioned by the VLM, leading ReSafe to fail in removing the harmful content.

**Questions:**

1. It is suggested to discuss the computation cost of ReSafe and potential way to improve efficiency.

2. It is suggested to enhance the evaluation of how well ReSafe preserves safe semantic and visual information.

3. It is suggested to discuss the proposed method's robustness against adversarial attacks.

---

### Official Review · Reviewer_FisM · 2025-10-31

**Soundness:** 2
**Presentation:** 2
**Contribution:** 2
**Rating:** 2
**Confidence:** 4

**Summary:**

This paper addresses the vulnerability of T2I models to safety circumvention via adversarial prompts or concept arithmetic. The authors propose ReSafe, a post-hoc I2I translation framework that takes an unsafe image as input and regenerates a safe version. The method first extracts multimodal safe features by using a VLM to create safe and unsafe captions, which guide the removal of unsafe components from the image's visual embedding. It then uses Interpolated Prompt Optimization to combine these safe visual and textual features into a new discrete prompt. Finally, this safe prompt is fed to a standard T2I model to generate the final safe image.

**Strengths:**

1. The paper frames the T2I safety problem as a post-hoc I2I translation task. This is an original perspective, even if its practical utility is low.
2. The presentation of the paper is clear and easy to follow.
3. The Interpolated Prompt Optimization is interesting. By using Slerp to combine safe visual features and safe text features, the method effectively mitigates the information loss that often occurs in image-to-prompt pipelines that rely only on visual features.

**Weaknesses:**

1. The paper's core motivation is its greatest weakness. In any real-world deployment, if a service provider's system detects that an unsafe image has been generated (which is a necessary first step for ReSafe to even be activated), the standard, safest, and most logical response is to block the output entirely. The paper explicitly mentions that default safety filters "often return black images" and presents ReSafe as a better alternative because it "retains the utility of safe content". This justification is unconvincing. There is little practical incentive for a provider to expend massive computational resources to "fix" a harmful image when it can (and should) simply refuse to serve it. This makes the entire problem formulation feel artificial.
2. The ReSafe pipeline is extremely computationally expensive and slow. It requires a second full forward pass of a T2I model to regenerate the final image. The authors report this takes 118 seconds per image. The paper then suggests combining this already-slow method with other T2I safety methods, creating a computational stack that is entirely impractical for any real-time application.
3. In Figure 5, several images generated by ReSafe to remove Nudity still clearly depict nude or semi-nude figures. This suggests the NudeNet classifier with its $0.45$ threshold is not a reliable metric for safety, and the method's claims of effective removal are overstated.

**Questions:**

1. Could the authors provide a more compelling justification for the practical utility of this method? Why is regenerating a safe image preferable to simply blocking a detected unsafe image?
2. Given the extreme computational cost, what is the intended practical application for ReSafe? How can this latency be justified, especially when proposed as an add-on to other T2I safety methods?
3. Why was the much simpler baseline of using an LLM to rewrite the original unsafe text prompt into a safe one not included in the experiments? This seems like a more direct, efficient, and practical solution to the same problem.
4. What is the formal definition of "unsafe" or "harmful" used in this paper? The evaluation relies on automated classifiers and subjective categories (nudity, violence, artist styles ), but a precise, formal definition of the concepts being removed is not provided.

---

### Note · Authors · 2025-11-13

I have read and agree with the venue's withdrawal policy on behalf of myself and my co-authors.